# Seismic Indirect Economic Loss Assessment and Recovery Evaluation Using Night-time Light Images—Application for Wenchuan Earthquake

Jianfei Wang[1], Jingfa Zhang[2], Lixia Gong[2], Qiang Li[2], Dan Zhou[3]

[1]Institute of Engineering Mechanics (Key Laboratory of Earthquake Engineering and Engineering Vibration), China Earthquake Administration, Harbin 150080, China
[2]Institute of Crustal Dynamics, China Earthquake Administration , Beijing 100085, China
[3]Institutes of Science and Development, Chinese Academy of Science, Beijing 100190, China

*Correspondence to*: Jingfa Zhang (zhangjingfa@hotmail.com)

**Abstract.**Seismic indirect economic loss is not only impact regional economic recovery policies, but also related to the economic assistance at the national level. Due to the Cross-regional economic activities and the difficulty of obtaining data, it's difficult that the indirect economic loss survey covers all economic activities. However, night-time light in an area can reflect the economic activity of the region. This article focuses on the indirect economic losses caused by the 2008 Wenchuan earthquake and evaluation of the progress of restoration and reconstruction based on night-time light Images. First, the functional relationship between GDP and night-time light parameters is established based on the pre-earthquake data. Next, the indirect loss of the earthquake is evaluated by the night-time light attenuation in the disaster area after the earthquake. Then, the capacity recovery, which is characterized by the brightness recovery process of the light area, was evaluated. Lastly, the process of light expansion in the disaster area was analyzed to evaluate the speed and efficiency of economic expansion.

## 1 Introduction

Following the accelerated pace of global economic integration and the rapid growth of population and social wealth, the damage caused by earthquake disasters is characterized by large amounts, far-reaching effects, and long recovery periods (Pielke et al., 2008). Economic recovery and reconstruction are important targets for post-earthquake economic policies (Lyles et al., 2014). However, due to the lack of detailed post-disaster economic recovery tracking data, economic policy plan is divided based on the amount of direct economic losses which result in insufficient policy sustainability (Song et al., 2017). When planning the allocation of post-disaster aid funds and medium- and long-term economic policies, decision makers mainly rely on the current market value of production data in the disaster area and the reconstruction cost of the production environment. Insufficient estimates of indirect losses are likely to result in gaps between aid funds and actual demand, even overlooking relatively remote disaster areas (Ge et al., 2010). Therefore, it is important for post-disaster macroeconomic policies and earthquake to insurance that indirect economic losses and economic recovery in the affected areas are accurately assessed.

The 2008 Wenchuan earthquake(Ms8.0) is the biggest earthquake event in China since 1970s. This earthquake struck Sichuan Province, China on May 12, 2008. It killed nearly 70,000 people, 18,000 people were missing, and more than 370,000 people were injured. Millions of people were made homeless by the quake, the cost of which was estimated at $86 billion (Kenneth et al.,2013). In the three years after the disaster, Chinese government spent 865.8 billion yuan to complete 29,692 aid projects, which has brought Chinese Power to the attention of the world (Gu, 2018).

There are many researches about the recovery of the disaster area after the Wenchuan earthquake, which can be generally classed as three categories. Firstly, researches were focused on the assessment vegetation and environmental carrying capacity in the disaster area. For instance, Zhao et al. (2009) evaluated the soil loss after the Wenchuan earthquake and converted the losses into monetary values based on the environmental economics principles. Zhao et al. (2014) and Yang et al. (2018) assessed the restoration of vegetation in the affected areas of the Wenchuan earthquake by comparing remote sensing data with GVG (GPS、Video & GIS) agricultural sampling data. Secondly, researches concentrated on the evaluation of the disaster area macroeconomic losses. For example, Zhu et al. (2018) assessed the seismic economic losses based on the GDP (Gross Domestic Product) growth model, Wu et al. (2012) considered the impact on activities within the economic activity system after the earthquake and then assessed the indirect economic losses of the Wenchuan earthquake by the adaptive regional input-output model (ARIO). The third is the assessment of post-disaster community and social resilience. Liu et al. (2018) assessed the restoration of buildings and infrastructure in the Wenchuan disaster area by remote sensing and actual interview data, and Le et al. (2017) assessed the perception of recovery (PoR) of the Wenchuan earthquake-affected area from house recovery condition (HRC), family recovery power (FRP) and reconstruction investment (RI) based on a structural equation model. In a word, the current post-disaster economic monitoring is mainly based on manual surveys and regional statistical data, which are inefficient and have a large spatial scale, while remote sensing (RS) and geographic information systems (GIS) have significant advantages in the fineness of measurements and spatial distribution of post-disaster loss and recovery assessments.

Because of the obvious advantages of periodic economic monitoring (Zhou et al.,2015; Li et al.,2017;Tan,2017), night-time light has been widely recognized in the field of regional economic monitoring (Li and Li,2015; Fu et al.,2017). The application of night-time light in earthquake disasters is mainly divided into two parts. In the first part, night-time light can be used to identify earthquake-affected areas and to assess disaster losses. The night-time light was first applied to identify earthquake-affected areas in the 1999 Marmara earthquake. Hashitera et al. (1999) evaluated the impact of earthquakes in the disaster area based on a series of DMSP-OLS data, which contain pre-earthquake and post-earthquake night-time light images. Fan et al. (2018) determined the disaster level and area of the disaster based on the brightness changes of VIIR night-time light images from 3 months before earthquake to 3 months after earthquake in the disaster area. Kohiyama et al. (2004) developed the EDES disaster-stricken area determination system, which is based on the DMSP-OLS night-time light images, to delineate the earthquake-affected areas quickly within 24 hours after the disaster. In the second part, night-time light can be used for post-disaster economic recovery monitoring. Zhang (2018) focused on the relationship between night-time light and the number of deaths, missing persons, and building collapse rates in the earthquake-affected areas of the

Wenchuan earthquake in 2003-2013. Gillespie et al. (2014) tracked the changes of night-time light from 2004 to 2008 and then studied the relationship between the brightness changes and some indicators such as per capita consumption, energy consumption and economic recovery capacity at the community level, thus evaluating the recovery after earthquake in Indonesia. Andersson et al. (2015) evaluated the recovery of post-disaster economic activity in southern Thailand by monitoring the recovery of night-time light in South-East Asia from 2005 to 2006. The School of Economics and Finance (2018) also assessed the long-term economic impact of the New Zealand earthquake based on the changes of night-time light intensity. It has been widely proven that there is a close relationship between night-time light and economic activity in the disaster area. We believe that the changes of night-time light after earthquake can reflect the earthquake production capacity of the disaster area and can therefore be used for seismic indirect economic loss assessment.

## 2 Data and Processing

In terms of recovery and reconstruction, it is very important to understand the indirect economic loss and recovery assessment of the Wenchuan Ms 8.0 earthquake, which is a significant earthquake in recent years. This study chose Sichuan Province as the research area. In order to avoid the impact of the 2013 Lushan earthquake, this paper focuses on the indirect economic impact of the Wenchuan earthquake on Sichuan Province from 2008 to 2012.

### 2.1 Data sources

The GDP data were obtained from Sichuan Provincial Bureau of Statistics (http://www.sc.stats.gov.cn/), which publishes the *Sichuan Statistical Yearbook* every year. The night-time light obtained from the National Oceanic and Atmospheric Administration (NOAA, http://www.noaa.gov) and the night-time light images (Figure 1) obtained by Operational Linescan System (OLS) of Defense Meteorological Satellite Program (DMSP) from 2000 to 2012 were used in this study. The satellite altitude of DMSP is approximately 830 km, its swath is 3000 km, and revisit cycle is approximately 101 minutes, and the satellite can orbit the earth 14 times a day and obtain 4 global coverage maps. DMSP/OLS can be used to detect human activities such as town lighting, aurora, lightning, fishing, fire, etc. The NOAA provides three types of night-time light data. 'cf_cvg' is the total amount of brightness observation under cloud-free conditions, 'avg_vis' is the average observed brightness under cloud-free conditions, and 'stable_lights.avg_vis' is the average brightness of the stable light source under cloud-free conditions. This paper selected the average brightness of the stable light source under cloud-free conditions.

### 2.2 Radiation correction

The DMSP/OLS night-time light images during 2000-2012 came from the F15, F16 and F18 satellite. Due to the differences in satellite and other imaging conditions, radiation correction is required before quantification of night-time light image of different times.

This paper used the invariant region method proposed by Cao (2015) who gives the logistic radiation correction model of the Chinese region's night-time light image. All images were calculated following Eq. (1):

$$DN_{cal} = a \times DN^b,$$ (1)

where $DN_{cal}$ is the corrected image value, DN is the original image value to be corrected, and a and b represent two different constants. The correction effect is shown in Figure 2.

The radiation correction greatly increases the smoothness from 2003 to 2004 and from 2010 to 2011; it indicates that the radiation correction is very effective.

**3 Method**

The seismic direct economic loss refers to the damage of existing production materials and environment by earthquake, which mainly reflects the impact of earthquake disasters on economic stock. However, the seismic indirect economic loss is a systematic manifestation of losses in the chain of economic activities, which focuses on the far-reaching impact of disasters on economic flows. On the one hand, there are many studies show that night-time light can be used as a proxy of economic activity. Chen et al.(2011) have proven nighttime luminosity could be used to improve estimates of output at the regional level, Bruederle et al.(2018) conclude that nighttime lights are a good proxy for human development at the local level, Ma et al.(2014) had proved that nightlight data could be indicative of demographic and socioeconomic dynamics in China's cities. Therefore, this paper proves that the changes in post-earthquake night-time light can reflect changes in the regional economic system. On the other hand, in order to ensure the fairness of disaster relief assistance, government should avoid the problem where the benefits of developed regions will cover up the economic difficulties of backward regions by transfer payment system. This paper defines the light recovery and lighting expansion in the disaster area as two different processes. The *Economic recovery evaluation model* only take increasing light intensity of the disaster area with light before the earthquake. And the *Economic expansion evaluation model* only take increasing light intensity of the disaster area without light before the earthquake. The technical flowchart is shown in Figure 3:

Firstly, the pre-earthlight image is clipped through administrative areas, counting the DN-related parameters in each administrative area, then its correlation with GDP is analyzed, and the parameter with the highest correlation is selected as the BF (best feature). Secondly, establishing the function of BF and GDP, differentiating the function, then the function of ΔBF (the change of BF) and ΔGDP (the change of GDP) is obtained. Thirdly, compared pre-earthquake and post-earthquake night-time light image, counted the ΔDN (the change of DN) in each administrative area, the indirect economic losses are evaluated by the function of step 2. Lastly, the areas where the post-earthquake DN is higher than pre-earthquake are extracted, the expansion areas are where the pre-earthquake DN equal to 0, and the recovery areas are where the pre-earthquake DN higher than 0.

## 3.1 Seismic indirect economic loss assessment model

The pre-earthquake night-time light images were clipped by the administrative divisions of Sichuan Province; then, the statistics of every administrative district's DN parameters, which include night lighting area, total brightness of light, average brightness of lighting area, and average brightness of administrative area, were collected. Through the correlation analysis of characteristic parameters and GDP, the most relevant characteristic parameter, which was defined as the parameter BF, was extracted. This established the function F(BF) as Eq. (2):

$$GDP = F(BF), \tag{2}$$

where GDP is the gross domestic product of each pre-earthquake administrative region, BF is the most relevant parameter. It is assumed that the productivity conversion efficiency, which can be regarded as the integral function of F(BF), is relatively close in adjacent areas. Therefore, the integral function of F(BF) can be used to calculate the GDP changes caused by the change of the annual total lights in each administrative district. It can be expressed as Eq. (3):

$$\triangle GDP = F'(BF) \cdot \Delta BF \tag{3}$$

## 3.2 Economic recovery evaluation model

It is easy to ignore those areas with weak economic foundations, it may make reconstruction funds concentrated in the areas with higher economic levels. However, this approach is not in line with the principle of fairness in disaster assistance. It is assumed that the night-time light of the human activity area will not disappear under the normal conditions, and the pixel whose DN value decreases after earthquake is the area to be restored. When the brightness of the pixel reaches the level of pre-earthquake, it can be considered that the economic level of the area has returned. The earthquake disaster economic recovery assessment model is as Eq. (4) and Eq. (5):

$$YR_n = \frac{\sum_{i=1,j=1}^{i=\max(i),j=\max(j)} R_n}{\sum_{i=1,j=1}^{i=\max(i),j=\max(j)} PR_{t0}}, \tag{4}$$

$$AR_m = \sum_{n=1}^{n=m} YR_n, \tag{5}$$

where $AR_m$ is the proportion of the cumulative recovery area in $m$ years after the earthquake, $YR_n$ is the proportion of recovery area in $n$-th year after the earthquake, $R_n$ is the matrix of recovery area in $n$-th year after the earthquake, $PR_{t0}$ is the damaged area, $i$ is the rows of a matrix, and $j$ is the column of the matrix. Therefore, $R_n$ and $PR_{t0}$ are Boolean matrices and can be calculated following Eq. (6) and Eq. (7):

$$PR_{t0(i,j)} = \begin{cases} 1, & DN_{t0(i, j)} > 0 \\ 0, & DN_{t0(i, j)} = 0 \end{cases}, \tag{6}$$

$$R_{n(i,j)} = \begin{cases} 1, & PR_{t0(i,j)} * DN_{t0+n(i,j)} > DN_{t0(i,j)} \\ 0, & PR_{t0(i,j)} * DN_{t0+n(i,j)} = DN_{t0(i,j)} \end{cases}. \tag{7}$$

The $PR_{t0(i,j)}$ in equation 6 refers to the value of the $i$ row and $j$ column in $PR_{t0(i,j)}$. $DN_{t0}$ is the pre-earthquake night-time light. In this paper, the value of t0 is 2007, and $DN_{t0(i,\ j)}$ is the value of the night-time light image. $R_{n(i,j)}$ in equation 7 refers to the value of the $i$ row and $j$ column in $R_n$, $DN_{t0+n}$ is the night-time light in $n$-th year after the earthquake, and the symbol * indicates that the corresponding elements in the two matrices should be multiplied.

## 3.3 Economic expansion evaluation model

The economic expansion of the earthquake-affected areas in the post-earthquake period was mainly manifested in two aspects. On the one hand, the earthquake destroyed the inefficient industrial structure and rebuilt a more advanced scientific industrial structure, which has led to the expansion of the regional economy. The *Overall Plan for Wenchuan Earthquake Recovery and Reconstruction* formulated by the Chinese Government requires that disaster reconstruction should focus on technological innovation, it should promote structural adjustment and transformation of development modes, and improve the self-development capacity of stricken areas. On the other hand, the needs of recovery and reconstruction expanded the market for many industries. According to the Statistics Bureau of Sichuan Province, the GDP of 9 stricken counties in 2014 (52.12 billion yuan) reached 2.32 times that of 2008 (22.47 billion yuan). This paper evaluated the economic expansion post-earthquake based on the new night-time light area. The evaluation model is as Eq. (8) and Eq. (9):

$$YE_n = |PR_{n-1} - 1| * (DN_n - DN_{n-1}) , \tag{8}$$

$$AE_m = \sum_{n=1}^{n=m} \sum_{i=1,j=1}^{i=\max(i),j=\max(j)} YE_n , \tag{9}$$

$YE_n$ is the night light growth matrix in $n$-th year after the earthquake, $|PR_{n-1} - 1|$ is the non-light area in $(n-1)$-th year after the earthquake, $DN_n$ is the value of night-time light in $n$-th year after the earthquake, $DN_{n-1}$ is the value of night-time light in $(n-1)$-th year after the earthquake, and $AE_m$ is total light expansion in $m$ years after the earthquake.

## 4 Results

### 4.1 Indirect economic loss of Wenchuan earthquake

The total pre-earthquake DN of Sichuan is the best parameter to establish a regression model with GDP. Firstly, the correlation between the total GDP and the DN-related parameters were analyzed (Figure 4), which include the lighting area, total DN, mean DN in light area, and mean DN in administrative in Sichuan. On the one hand, the light area and the mean DN in light area have an unstable correlation curve with GDP. During the post-earthquake period, the correlation between light area and GDP was decreasing, while the correlation between the mean DN in light area and GDP was increasing. On the other hand, the total DN and the mean DN in administrative have a stable correlation curve with GDP, and the total DN have a higher correlation with GDP than the mean DN in administrative. It is obvious that the correlation coefficient between the total DN and the GDP is the highest, which is basically greater than 0.95 in each year. Therefore, the total brightness is selected as the DN parameter. Secondly, the largest value of Chengdu was removed from the sample. And then, the scatter

plots of the two sets of data samples were compared (Figure 5), the blue point shows the pre-earthquake relationship between the total DN and GDP, and the orange point shows the post-earthquake relationship. It shows that the pre-earthquake relationship between the two parameters is strong and the post-earthquake relationship between the two parameters is poor. Third, established a regression function of GDP and total brightness (Figure 6), which is Eq. (10):

$$GDP = 4 \times 10^{-8} \times DN^2 + 0.0042 \times DN + 96.285 \tag{10}$$

The $DN$ in Eq. (10) is the total DN in the administrative. The Eq. (10) is differentiated based on the Eq. (3), then the economic losses can be expressed as Eq. (11):

$$\Delta GDP = (8 \times 10^{-8} \times DN + 0.0042) \times \Delta DN \tag{11}$$

The total DN of each administrative area after Wenchuan earthquake is shown in table 1, then the indirect economic losses of Sichuan Province after the Wenchuan earthquake is assessed by Eq. (11), the loss is shown in table 2.

## 4.2 Economic recovery progress in Sichuan Province

The brightness of the night-time light image of 2007 is used as a reference, when the brightness returns to the previous brightness, the scale is considered to be the economic recovery area (Figure 7). Yellow represents the area that recovering to the pre-earthquake level during the first year, green stands for the area recovering to the pre-earthquake level during the second year, cyan represents the area that recovering to the pre-earthquake level during the third year, purple represents the area that recovering to the pre-earthquake level during the fourth year, fuchsin represents the area that recovering to the pre-earthquake level during the fifth year, and red represents the area that have not yet recovered until the fifth year. On the one hand, the post-earthquake annual economic recovery areas were assessed (Figure 8) based on the proportion of the brightness recovery scales (ReS). The recovery area in Zhangzhou, Nanchong, Neijiang, Suining, Yibin, Zigong and Ziyang was small in 2008 but grew quickly in the next years. By contrast, the recovery area in other cities was larger in 2008 but grew slowly over the next years. While the recovery areas were about 60-70% of stricken areas, they were still not fully recovered in 5 years. On the other hand, the economic recovery was evaluated by the ratio of post-earthquake DN to pre-earthquake (DN_ratio) in regions (Figure 9). The economic in most of the recovery areas showed a significant retreat in 2009, and except for Dazhou, Zhangzhou, Aba, Panzhihua, Suining, Ya'an and Yibin, the other cities' economic was restored to 80% in 2012. In summary, our results indicate that the economic recovery in Sichuan province take more than 5 years, which is far more than the time spent on reconstruction (Yang,2012).

## 4.3 Expansion of new economic zone

The night-time light image of 2007 was defined as reference to assess the expansion of the new economic zone (Figure 10). Yellow represents the area that covered by the night-time light in pre-earthquake, orange represents the expansion area of night-time light in 2008, green represents the expansion area of night-time light in 2009, cyan represents the expansion area of night-time light in 2010, purple represents the expansion area of night-time light in 2011, red represents the expansion area of night-time light in 2012. First, the ExpDN (cumulative DN of expansion area) was used to measure the cumulative

capacity of new economic zone in post-earthquake (Figure 11), it shows that the cumulative capacity of cities is extremely uneven. The new economic area of Liangshan, Guangyuan and Ganzi had a relatively high cumulative capacity, while other cities are relatively low. The cumulative capacity of all new economic zones had increased significantly in 2010 and 2011, but it had declined since 2012. Second, the ExpAvDN (average DN of expansion area) was used to measure the capacity

utilization of new economic zone (Figure 12), and the higher ExpAvDN, the higher annual average economic output in the region, which is also called capacity utilization. It is obvious that the capacity utilization of cities in 2008 are similar, the capacity utilization of Ganzi, Guangyuan, Liangshan, Mianyang and Nanchong have increased and the other regions declined in 2009, the most of regions' capacity utilization increased again in 2011, and the capacity utilization of Ganzi, Guangyuan, Liangshan, Panzhihua and Ziyang are relatively high in 2012.In summary, the new economic zone in the south

and east, which have a high cumulative capacity, declined significantly in 2010 and then grew rapidly.

## 5 Discussion

Compared with statistical data, night-time light can not only reflect the spatial difference of economic expansion and recovery but also guarantee the accuracy of assessment. In terms of short-term economic loss, the economic loss of this model in 2008 is 95.7 billion yuan, which is closer to Sun's results (100.8 billion yuan), but lower than Lu's results (168

billion yuan). The reason why this difference exits is that although these models taken into account the capacity reduction caused by earthquake, Lu's model (2008) assumed that the affected department was completely shut down during a period of time, while this paper's model and Sun's model (2011) assumed that the production capacity was gradually restored. In terms of mid-long term economic loss, the economic loss of this model from 2008 to 2011 is 596.8 billion yuan, which is more than Wu's results (463.4 billion yuan), and the economic loss of this model from 2008 to 2012 is 709.6 billion yuan, and

whose results is closer to Sun's results (645.4 billion yuan).The reason for this difference is that although both models measured the impact of input changes on output after earthquake, Wu's ARIO model (2012) and Sun's Harrod-Domar model (2011) predicted the output based on government public expenditure, while this paper's model measured the output based on the total investment in society which could be reflected by night lights. Generally, the indirect economic losses assessment in this paper are close to most of the evaluation results in other papers (Figure 13).

The quantitative relationship between night-time light and economic statistics in the post-earthquake years is abnormal. It's the result of the joint action of economic suppression, which caused by the earthquake damage, and economic promotion, and the reconstruction of the disaster area. The reasons for the economic shocks in earthquake-stricken areas conclude two aspects. On the one hand, due to the bullwhip effect of disasters, the main impact of direct losses on the industrial chain will take longer to show up. Due to the suppression caused by the earthquake, the purchasing power of the victims was reduced.

In order to reduce losses, suppliers should try reduce inventory as possible. Further, suppliers were also affected by earthquake, they tended to conservative strategies, and the reduction was magnified. It takes 1 year for the changing information about supply and demand to be transmitted to the economic system. Then, the gap was magnified and caused a

turbulent change in the indirect economic loss in the next years. It explains why the economic recovery show a significant decline in 2009. On the other hand, relief and reconstruction will stimulate the market in the disaster area. With the end of the relief activities, the market development of the disaster area will fall to a certain extent. Lots of new economic zones have developed significantly in 2010. The areas with rapid economic expansion are mainly distributed in the western

Sichuan eco-economic zone, the northeast economic zone, and the Panxi economic zone. The pillar industries in these regions are mainly tourism, building materials and energy. The policy of reconstruction in 2008-2010 developed these industries in a short term. Then, the faster regions have gradually declined in 2012, which may be the result of the post-disaster recovery policy. China's reconstruction policy was basically finished (Gu, 2018) in 2010, and the demand for construction, energy and other industries was reduced. Therefore, the production capacity of the new economic zone in

Guangyuan, Liangshan and Panzhihua began to fall off after 2011.

The path of economic change in the disaster area after the earthquake gradually spread from the northeast to the southeast (Figure 14). The economy level for 5 years after the earthquake in disaster area (2008-2012) were compared to the level of 2007. We found that in the first year (Fig. 14-a), the economic decline is mainly concentrated in the disaster area, and the city's economic decline is serious. In the second year (Figure 14-b), the economic slowdown area began to spread to some

major cities in the southeast of the disaster area. It was not until the third year (Fig. 14-c) that the disaster area began to recover, and the first one was not the area where the economy was reduced seriously after the disaster, but some cities located near these areas in the northeast. The force gradually spread from north to south during 2010-2012 (Fig. 14-c, Fig. 14-d, Fig. 14-e). Eventually, some cities were developed farther outside the disaster area (Figure 14-e). This explains the path of the Chinese government's aid funds from the north to the southwest and its radiation effects on the surrounding areas.

The area along the road or around the city turned around firstly (Figure 15). Previous analysis had proved that the economic recovery in disaster area began around 2010. Therefore, the process of economic recovery can be studied by comparing the economic level of the disaster area from 2009 to 2012 with the level of 2008. In 2009 (Figure 15-a), excepted for few areas along several roads close to the disaster area, there was only a small increase elsewhere. However, in 2010 (Fig. 15-b), the economy of the areas along most roads showed significant growth, and the surrounding areas of cities, which were seriously

affected by earthquake, showed an increasing trend. By 2011 (Fig. 15-c), the trend of economic growth in the disaster areas has spread to neighboring cities, and the urban economy has also begun to recover from the periphery to center. As of 2012(Fig. 15-d), the economy of disaster area showed an interesting development pattern. On the one hand, there were few recovery areas in the urban economy with severe recession; on the other hand, a series of significant growth were shown in the surrounding areas. This phenomenon is helpful for us to distinguish the industrial layout of the disaster-stricken areas

after the disaster. The secondary industries such as construction and manufacturing took a dominant place in the economically active areas after earthquake, while the tertiary industry such as service and entertainment industry took a dominant place in those  slowly recovery areas.

## 6 Conclusion

Due to the uneven distribution of aid funds and the adjustment of regional development strategies, the model of economic recovery and expansion differed from those before earthquake in study area. Using the night-time light data of DMSP/OLS, the indirect economic losses within 5 years after the Wenchuan earthquake were evaluated. Then the economic recovery progress of Sichuan Province was evaluated. The results are showed as follows:

(1) The GDP has a quadratic function relationship with the total night-time lights under normal conditions, and the indirect economic loss obtained by the function is consistent with the results obtained by economic statistical methods.

(2) The path of economic change in the disaster area after the earthquake gradually spread from the northeast to the southeast, meanwhile, it takes 1-2 years to show the main impact of earthquake on the economic system. Therefore, the economy of the disaster area after the earthquake reveal unstable and turbulent. Tourism, building materials, and the energy industry will experience a short-lived expansion after the earthquake, then a significant decline will be shown in the 3-4 years following the disaster.

(3) The economy of the areas along the road or around the city turned around firstly, and the economic recovery time is longer than the planned 5 years. Compared with the new economic zone, the capacity recovery of the earthquake site is slower. The recovery area is only approximately 60% of affected areas, and the production capacity can be restored to approximately 80% in 5 years.

In summary, night-time light has significant advantages in long-term economic monitoring in earthquake-stricken areas. It observe the geographical differences in economic development and growth mode in earthquake-stricken areas, and it also provides basis for macro-control of earthquake recovery and reconstruction.

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

**Table 1: The total DN in cities of Sichuan province after earthquake**

| ID | City | 2007 | 2008 | 2009 | 2010 | 2011 | 2012 | ID | City | 2007 | 2008 | 2009 | 2010 | 2011 | 2012 |
|---|---|---|---|---|---|---|---|---|---|---|---|---|---|---|---|
| | Total | 846204 | 761763 | 602885 | 721594 | 817014 | 771711 | 11 | Meishan | 28079 | 26228 | 18814 | 24423 | 29905 | 24832 |
| 1 | Bazhong | 10502 | 9288 | 4599 | 6724 | 10373 | 14969 | 12 | Mianyang | 57980 | 59248 | 47389 | 57527 | 63771 | 55599 |
| 2 | Chengdu | 230871 | 209496 | 181504 | 189890 | 203892 | 202837 | 13 | Nanchong | 41097 | 31762 | 22988 | 31281 | 41871 | 39038 |
| 3 | Dazhou | 39537 | 33282 | 20982 | 26989 | 31272 | 30842 | 14 | Neijiang | 23431 | 18967 | 16941 | 17800 | 23513 | 23050 |
| 4 | Deyang | 59965 | 56956 | 49748 | 54412 | 62130 | 56456 | 15 | Aba | 31961 | 26694 | 21717 | 30565 | 30281 | 27303 |
| 5 | Ganzi | 13617 | 14098 | 9879 | 13563 | 18214 | 17726 | 16 | Panzhihua | 53442 | 47699 | 33804 | 42436 | 40926 | 38390 |
| 6 | Guangan | 21679 | 19403 | 11776 | 17179 | 21012 | 20469 | 17 | Suining | 32384 | 25222 | 19535 | 20638 | 26588 | 26339 |
| 7 | Guangyuan | 18934 | 20453 | 18778 | 33606 | 32537 | 28184 | 18 | Yaan | 21678 | 21089 | 13722 | 17331 | 16252 | 13925 |
| 8 | Leshan | 29599 | 25883 | 20609 | 26260 | 28833 | 26644 | 19 | Yibin | 25632 | 20324 | 16570 | 18525 | 21971 | 18832 |
| 9 | Liangshan | 49065 | 52089 | 34013 | 46186 | 53709 | 50522 | 20 | Zigong | 13334 | 10583 | 10137 | 13140 | 13811 | 12950 |
| 10 | Luzhou | 21684 | 17118 | 15236 | 17096 | 19900 | 18210 | 21 | Ziyang | 21731 | 15882 | 14145 | 16023 | 26255 | 24594 |

**Table 2: Indirect economic loss in cities of Sichuan province (units: 100 million)**

| ID | City | 2008 | 2009 | 2010 | 2011 | 2012 | ID | City | 2008 | 2009 | 2010 | 2011 | 2012 |
|---|---|---|---|---|---|---|---|---|---|---|---|---|---|
| | Total | -956.57 | -2493.50 | -1612.87 | -904.57 | -1128.68 | 11 | Meishan | -11.93 | -59.72 | -23.56 | 0.00 | -20.93 |
| 1 | Bazhong | -6.12 | -29.75 | -19.04 | -0.65 | 0.00 | 12 | Mianyang | 0.00 | -93.61 | -4.01 | 0.00 | -21.05 |
| 2 | Chengdu | -484.56 | -1119.13 | -929.03 | -611.61 | -635.52 | 13 | Nanchong | -69.90 | -135.59 | -73.50 | 0.00 | -15.41 |
| 3 | Dazhou | -46.06 | -136.62 | -92.39 | -60.86 | -64.03 | 14 | Neijiang | -27.12 | -39.42 | -34.20 | 0.00 | -2.32 |
| 4 | Deyang | -27.08 | -91.93 | -49.97 | 0.00 | -31.58 | 15 | Aba | -35.59 | -69.22 | -9.44 | -11.35 | -31.48 |
| 5 | Ganzi | 0.00 | -19.77 | -0.29 | 0.00 | 0.00 | 16 | Panzhihua | -48.68 | -166.44 | -93.28 | -106.08 | -127.58 |
| 6 | Guangan | -13.51 | -58.77 | -26.71 | -3.96 | -7.18 | 17 | Suining | -48.63 | -87.25 | -79.76 | -39.36 | -41.05 |
| 7 | Guangyuan | 0.00 | -0.90 | 0.00 | 0.00 | 0.00 | 18 | Yaan | -3.50 | -47.21 | -25.79 | -32.20 | -46.01 |
| 8 | Leshan | -24.40 | -59.05 | -21.93 | -5.03 | -19.41 | 19 | Yibin | -33.17 | -56.64 | -44.42 | -22.88 | -42.50 |
| 9 | Liangshan | 0.00 | -122.30 | -23.40 | 0.00 | 0.00 | 20 | Zigong | -14.49 | -16.84 | -1.02 | 0.00 | -2.02 |
| 10 | Luzhou | -27.10 | -38.27 | -27.23 | -10.59 | -20.62 | 21 | Ziyang | -34.74 | -45.05 | -33.90 | 0.00 | 0.00 |

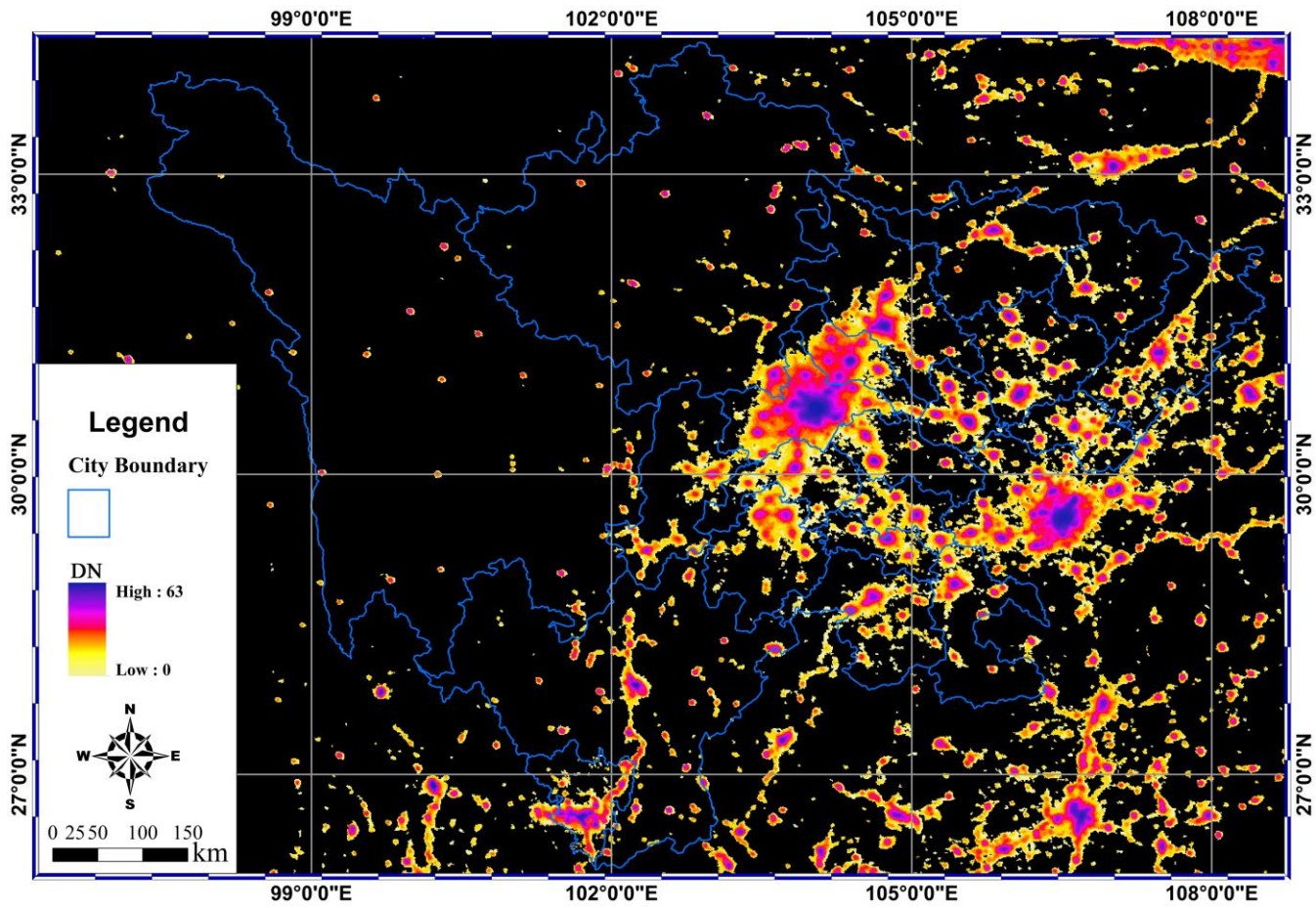

**Figure 1: The DN (digital number) of night-time light before the Wenchuan earthquake.**

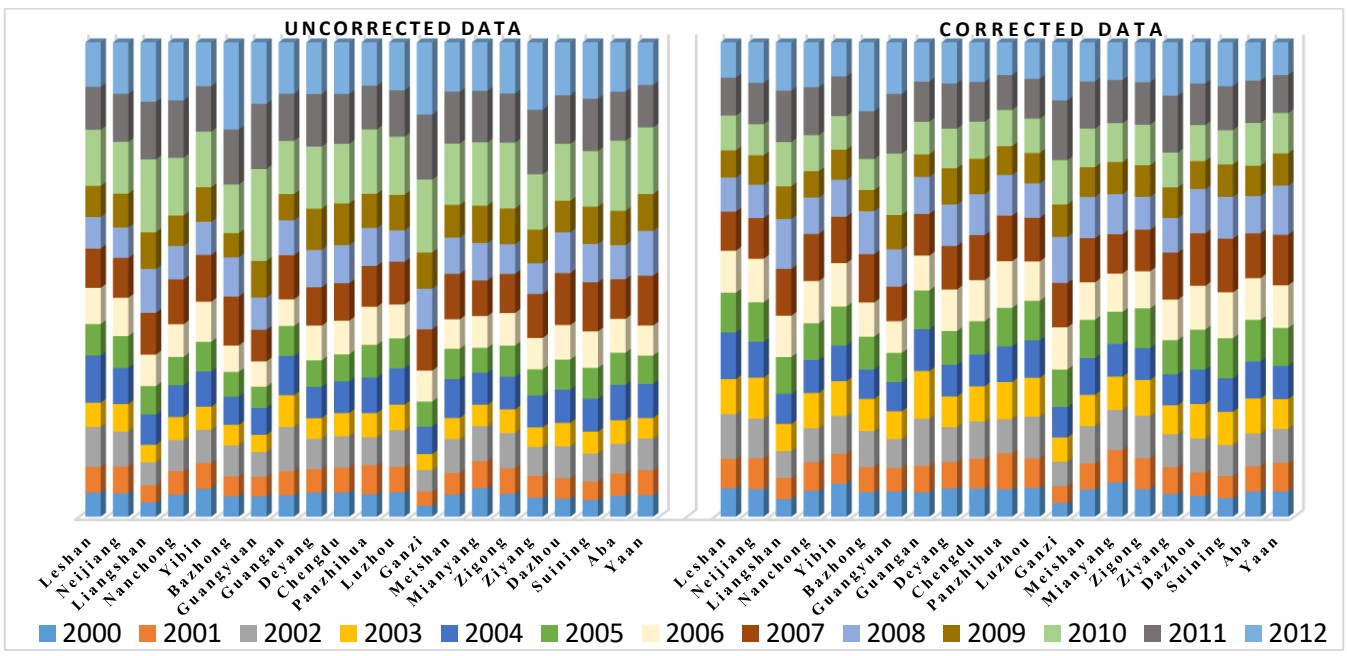

**Figure 2 Normalized total lightness of uncorrected and corrected data.**

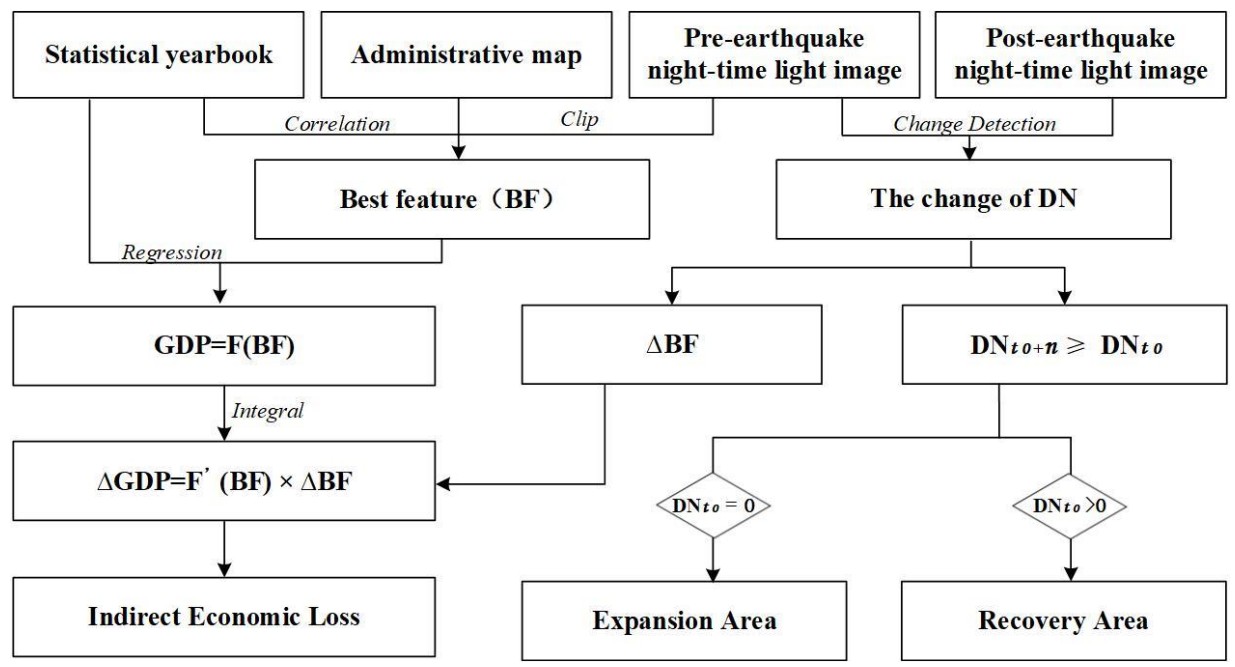

**Figure 3: Technical flowchart.**

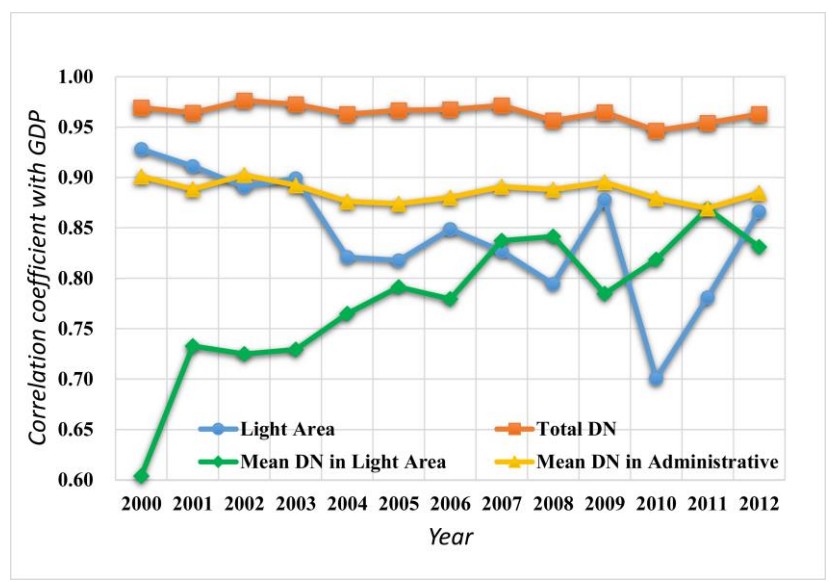

**Figure 4: Correlation coefficient between GDP and DN-related parameters.**

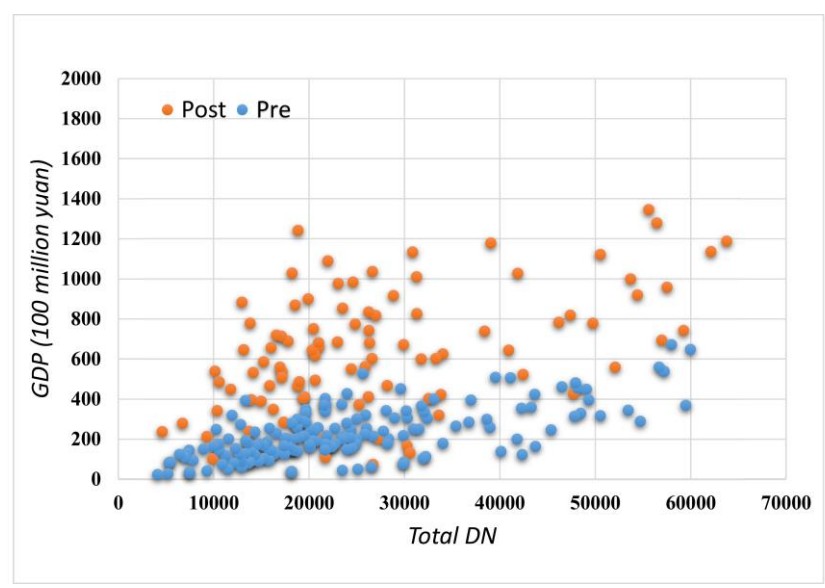

**Figure 5: DN-GDP scatter plots of pre- and post-earthquakes in Sichuan (Maxima have been excluded).**

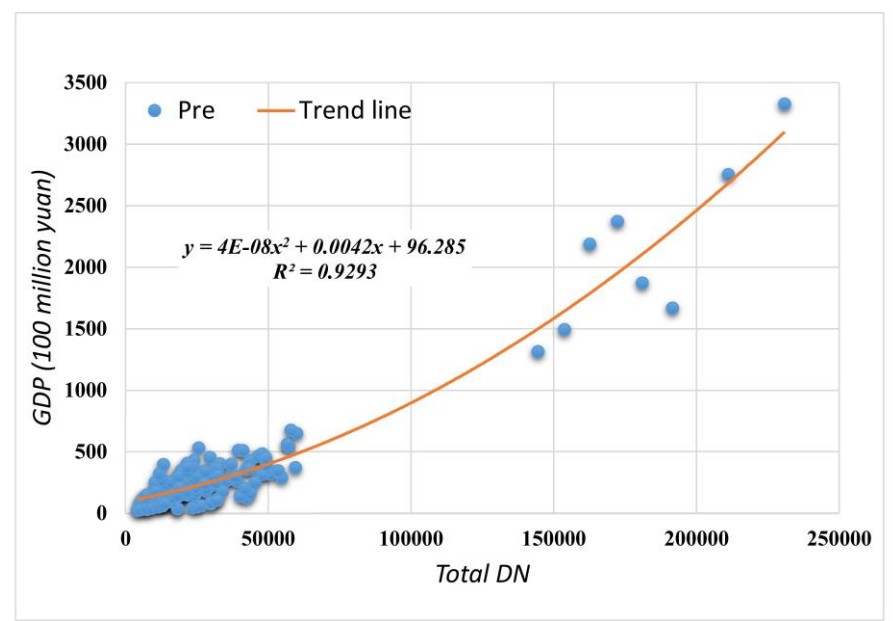

**Figure 6: Total DN and GDP Regression function of Pre-earthquake in Sichuan.**

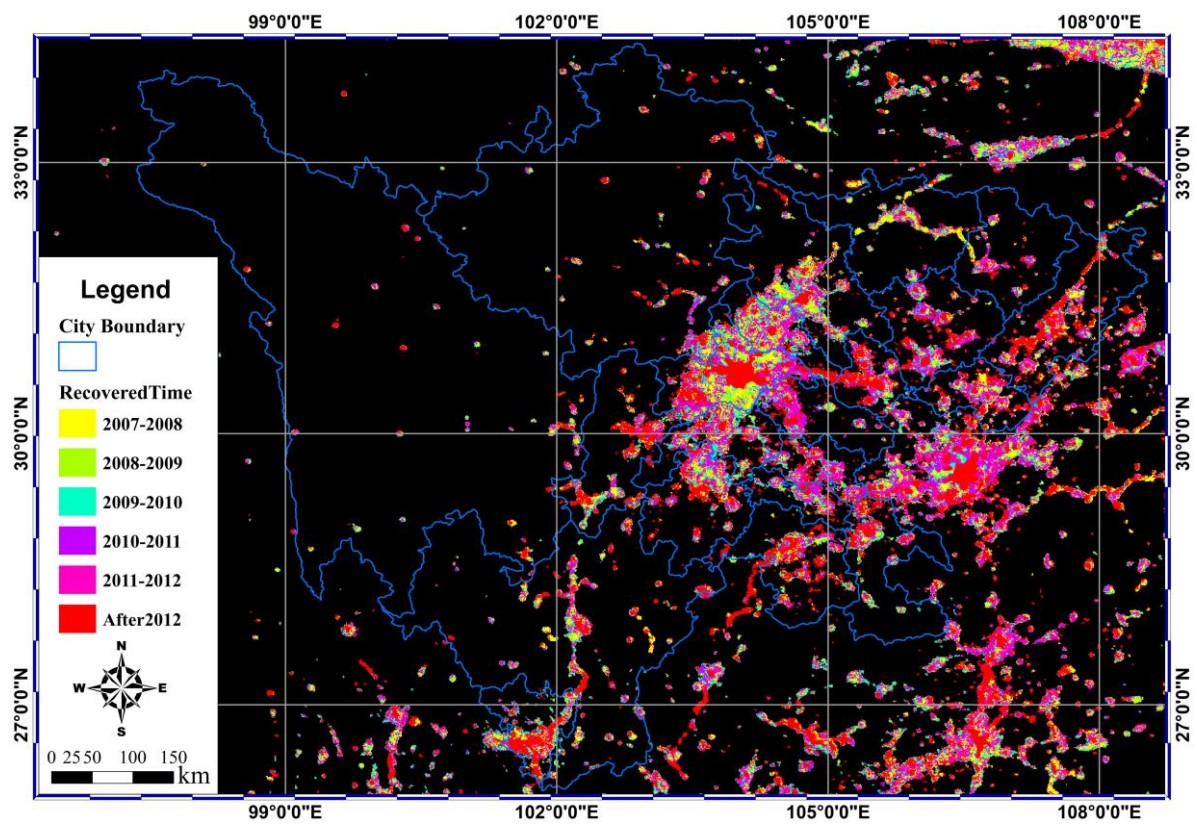

**Figure 7: The time of light recovery**

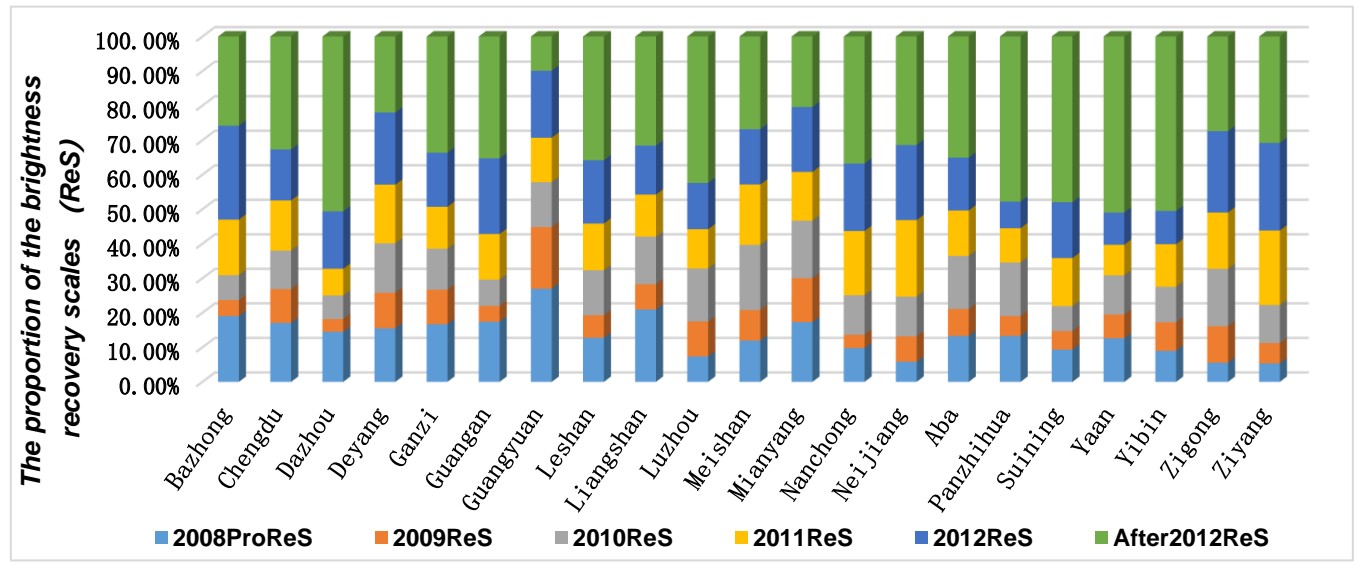

**Figure 8: The proportion of the brightness recovery scales.**

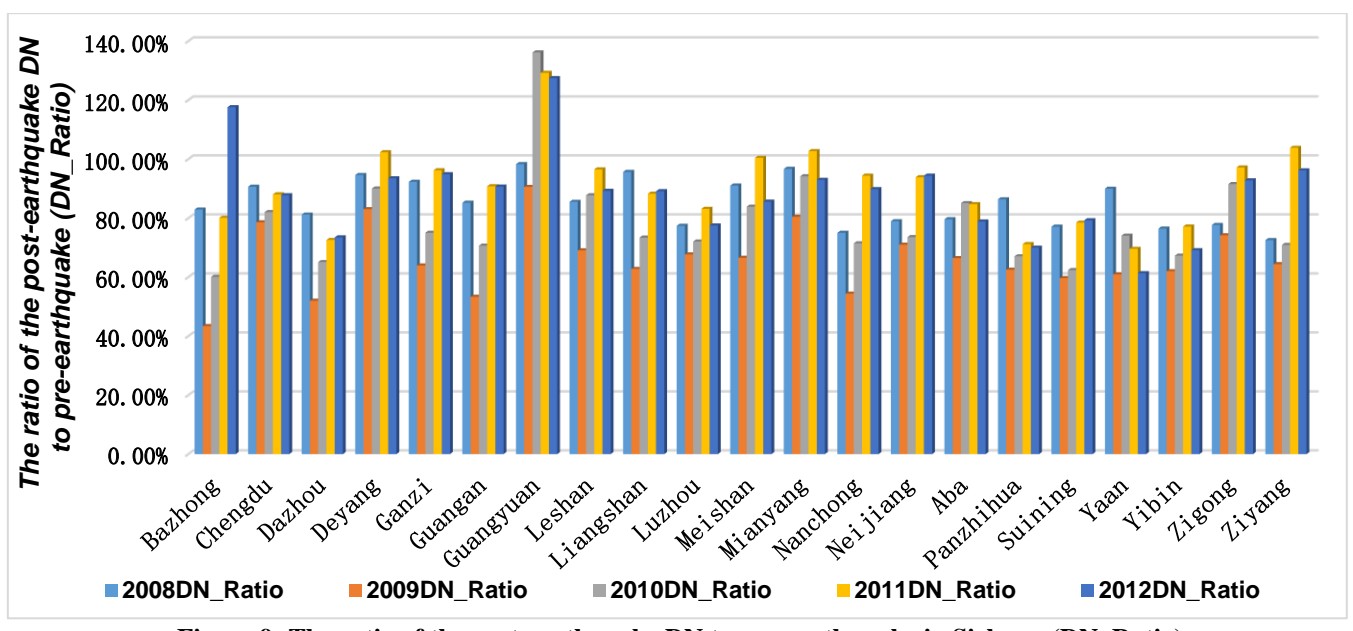

 **Figure 9: The ratio of the post-earthquake DN to pre-earthquake in Sichuan (DN_Ratio)**

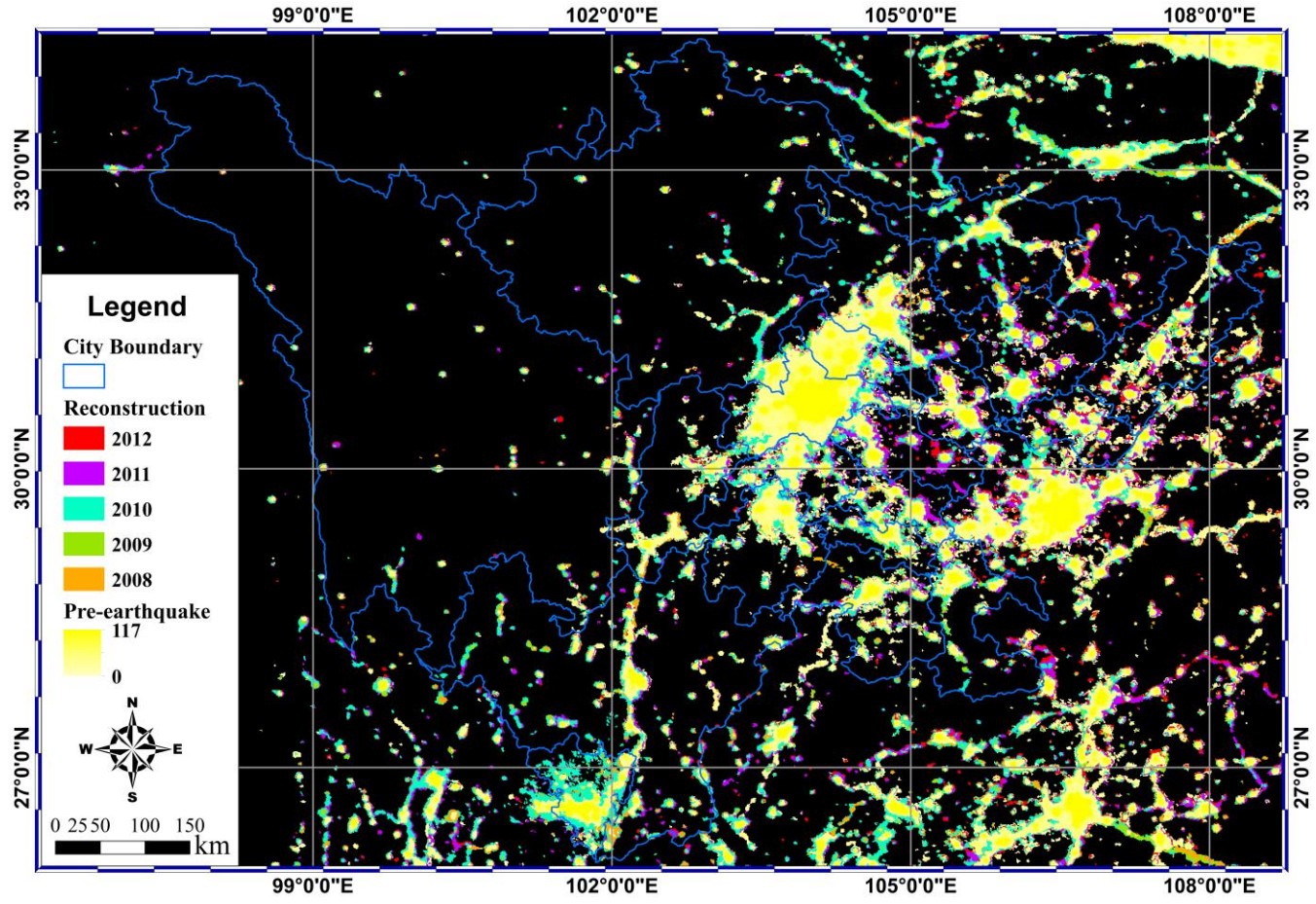

**Figure 10: Distribution of new lighting areas after earthquake**

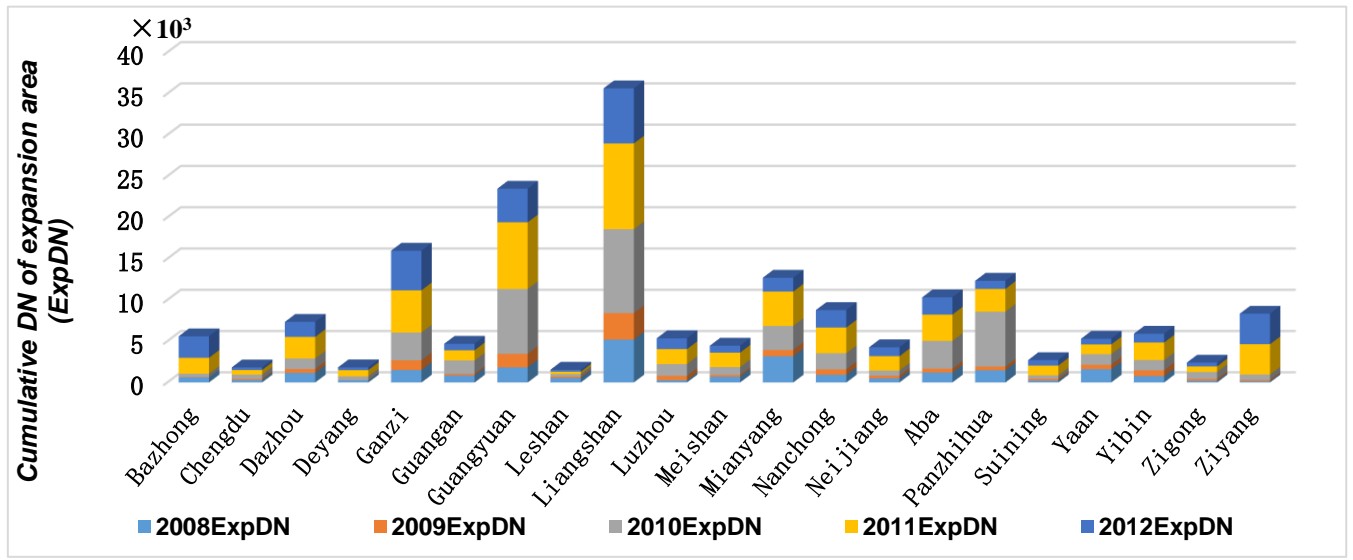

**Figure 11: The cumulative DN of expansion area**

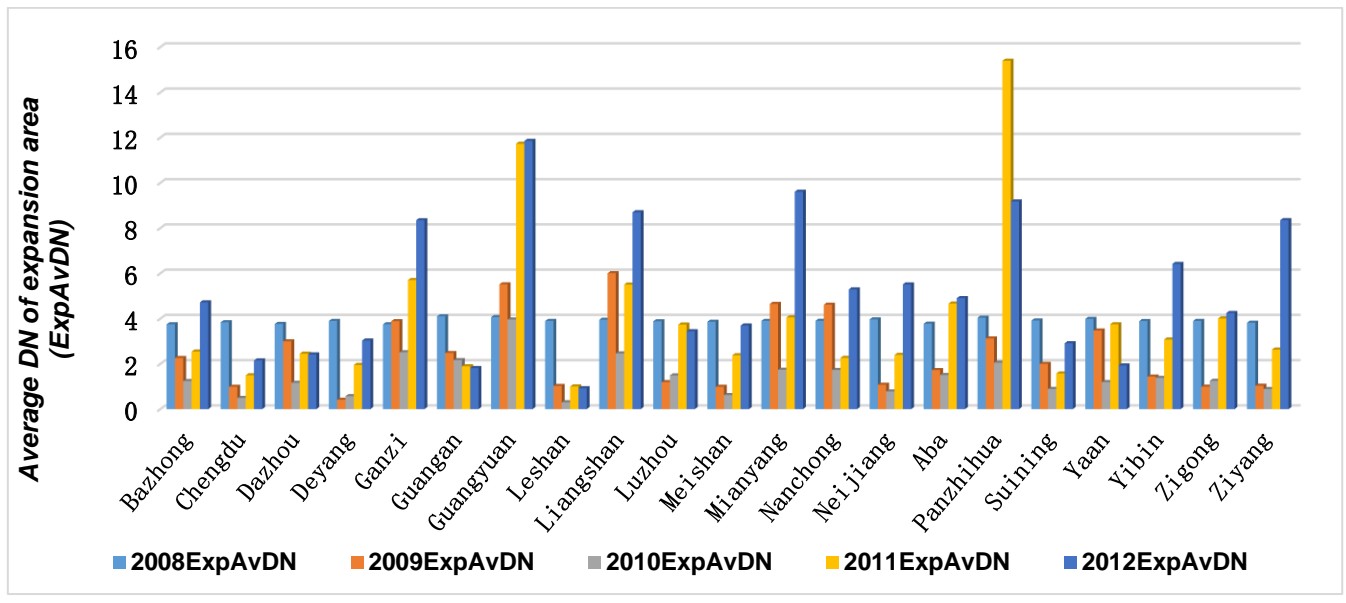

**Figure 12: The average DN in expansion area**

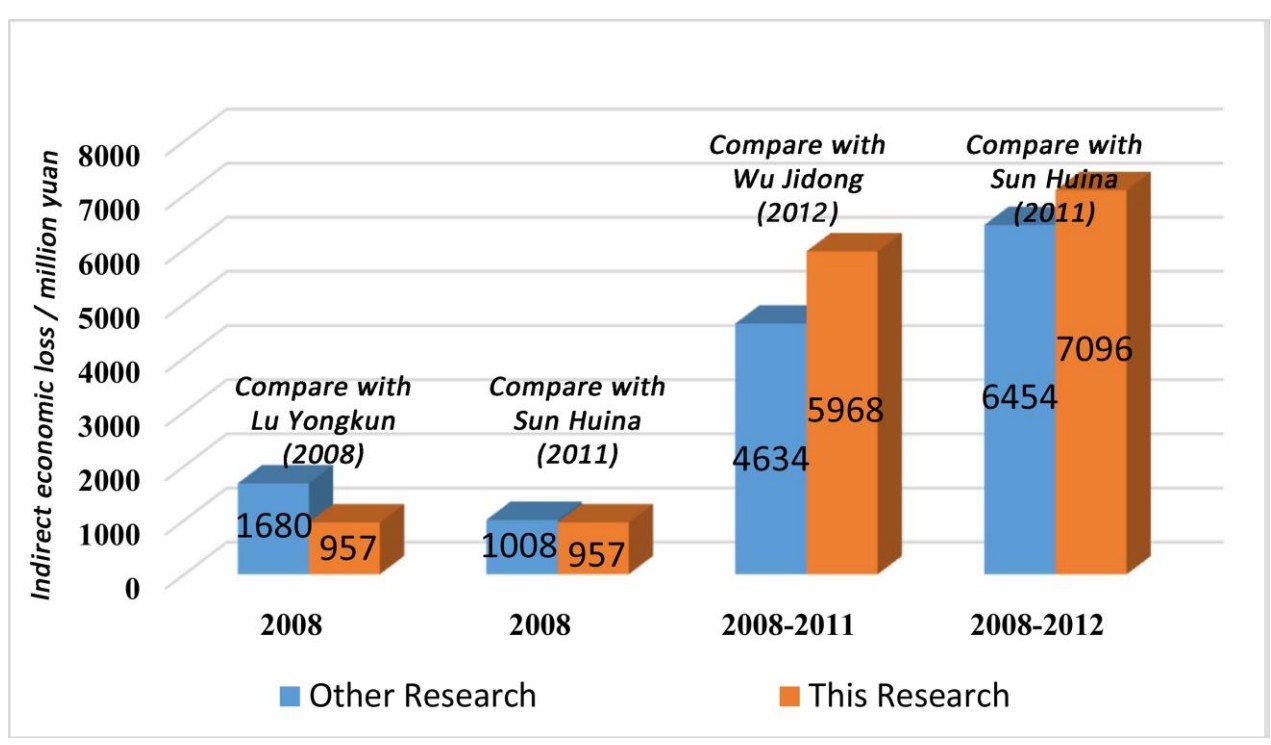

**Figure 13: Comparison of indirect loss with other research**

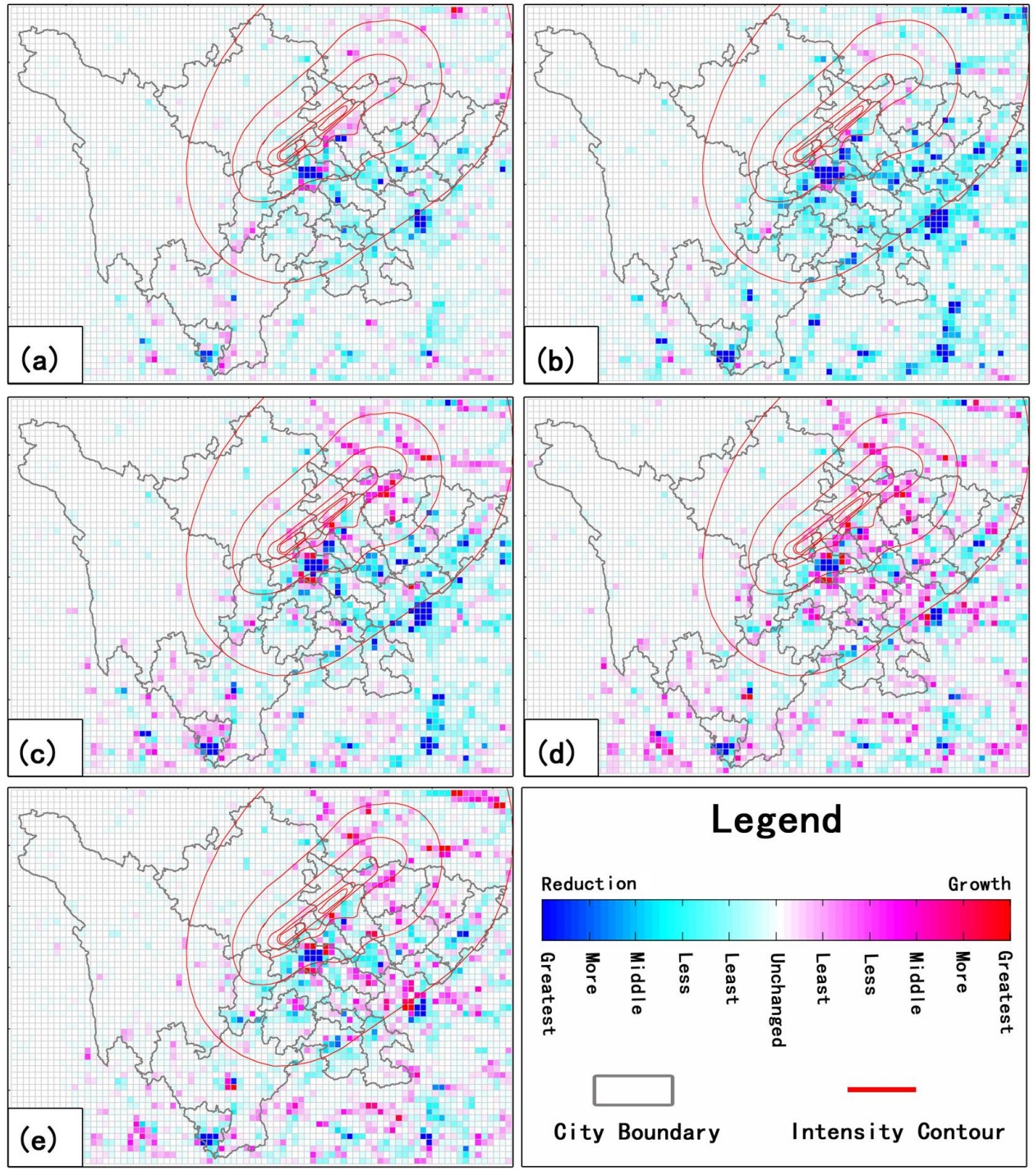

**Figure 14: The economic change path in disaster area ; (a) The path of 2008; (b) The path of 2009; (c) The path of 2010; (d) The path of 2011; (e) The path of 2012;**

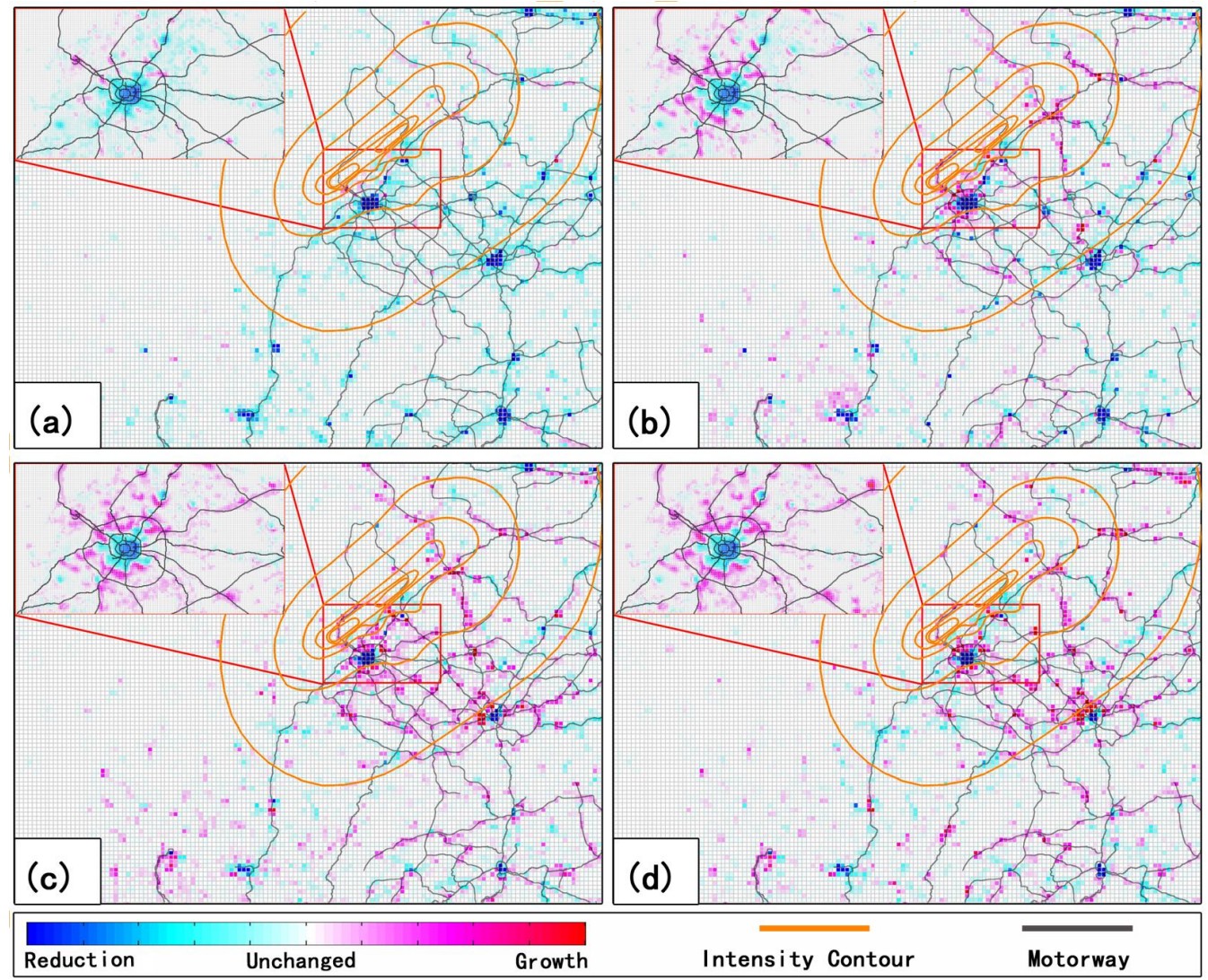

**Figure 15: Economic recovery process in the disaster area ; (a) Economic recovery area in 2009; (b) Economic recovery area in 2010; (c) Economic recovery area in 2011; (d) Economic recovery area in 2012;**