# Peer review of "Seismic Indirect Economic Loss Assessment and Recovery Evaluation Using Night-time Light Images—Application for Wenchuan Earthquake"

_Natural Hazards and Earth System Sciences, 2018_

## Referee Comment (RC1) · M. andersson (Referee) · 3 Sep 2018

The paper I have reviewed the manuscript which provide important insights to how remote sensing in particular night time light can be used to evaluate economic decline and economic recovery due to natural hazards. The manuscript is well written and suitable for publication in the journal Nat. Hazards Earth Syst. Sci..

I have a few comments for the authors to address in order to strengthen the manuscript: Comment 1 Citation from page 3 paragraph 5: "Although it has been widely proven

that there is a close relationship between night-time light and economic activity in the disaster area, due to the joint effect of economic suppression, which was caused by the earthquake damage, and economic promotion, which was caused by the reconstruction of the disaster area, the quantitative relationship between night-time light and economic statistics in the post-earthquake years is abnormal" Why is the relationship abnormal? Please explain better your point to why it is abnormal.

Comment 2 Citation from page 3 paragraph 10: "It is very important for recovery and reconstruction to understand the indirect economic loss assessment and recovery assessment of the Wenchuan 8.0 Ms earthquake, which is a significant earthquake in recent years." Why is this specific earthquake important to investigate? The reader needs more information about the earthquake. The authors need to put this specific earthquake in a better context.

Comment 3 This comment is rather general and it refers to the modelling (at page 4, 3. Method 3.2 Economic recovery evaluation model and 3.3 Economic expansion evaluation model) used to assess the economic recovery. Does the models take both increasing light intensity at one certain place and secondly, the increased spread of light i.e. new light sources at places where there were no light prior to the earthquake? Please clarify this in simple terms for the reader.

Comment 4 4.2 Economic recovery progress in Sichuan Province. Please start the paragraph with for example: Our results indicate that it has taken more than 5 years... Now the paragraph can be interpreted as a common observation. Make sure that this type of statement are done in line with your empirical results.

Comment 5 5 Discussion Citation from Page 7 Paragraph 5-10. "When considering the short-term impact of the stoppage of production after the earthquake, the indirect economic loss in 2008 evaluated by Lu (2008) was 168 billion yuan, which is 176% of the results found in this paper. According to the changes of economic growth rate after the earthquake, Sun (2011) predicted that the indirect loss after the earthquake in

2008 was 100.8 billion yuan, which is 105% of the results found in this paper." Please clarify this statement. What is the 176% resp. 105% telling us? Are your model overestimating the recovery? This is unclear in the present format.

---

## Referee Comment (RC2) · M. andersson (Referee) · 11 Sep 2018

I have read the comments given by the authors and feel satisfied with the authors′ answer. I suggest that the answers are added to the manuscript then I view this manuscript ready to be published.

---

## Author Comment (AC1) · 11 Sep 2018

Thanks for your contribution to this paper. I have read your comments very seriously and responded as follows:

Reply to the Comment 1: The relationship between night-time light and economic statistics was given in Figure 5. We have a brief discussion of this issue on the page 6 paragraph 1:" Second, the largest value of Chengdu was removed from the sample, and then, the scatter plots of the two sets of data samples, which are pre-earthquake

data and post-earthquake data, were compared (Figure 5). It is obtained that the correlation between the two parameters is strong in pre-earthquake data and poor in post-earthquake data." In the next revision, we will further supplement the description of the issue based on your comments.

Reply to the Comment 2: The 2008 Wenchuan earthquake(Mw8.0) is the biggest earthquake event in China since 1970s. This earthquake struck Sichuan Province, China on May 12, 2008. It killed nearly 70,000 people, 18,000 people were missing, and more than 370,000 people were injured. Millions of people were made homeless by the quake, the cost of which was estimated at $86 billion (Kenneth et al.,2013). In the three years after the disaster, Chinese government spent 865.8 billion yuan to complete 29,692 aid projects, which has brought Chinese Power to the attention of the world(Gu, 2018).

Reply to the Comment 3: In order to ensure the fairness of disaster relief assistance, government should avoid the problem that the benefits of developed regions will cover up the economic difficulties of backward regions by transfer payment system. This paper defines the light recovery and lighting expansion in the disaster area as two different processes. The 3.2 Economic recovery evaluation model only take increasing light intensity of the disaster area with light before the earthquake (Eq.6 & Eq.7). And the 3.3 Economic expansion evaluation model only take increasing light intensity of the disaster area without light before the earthquake (Eq.8)

Reply to the Comment 4: Our results indicate that the economic recovery in Sichuan Province took more than 5 years, which is far more than the time spent on reconstruction (Yang,2012).

Reply to the Comment 5:Compared with statistical data, night-time light can not only reflect the spatial difference of economic expansion and recovery but also guarantee the accuracy of assessment. In terms of short-term economic loss, the economic loss of this model in 2008 is 95.7 billion yuan, which is closer to Sun's results (100.8

billion yuan), but lower than Lu's results (168 billion yuan). The reason for this difference is that although these models taken into account the capacity reduction caused by earthquake, Lu's model (2008) assumed that the affected department was completely shut down during a period of time, while this paper's model and Sun's model (2011) assumed that the production capacity was gradually restored. In terms of mid-long term economic loss, the economic loss of this model from 2008 to 2011 is 596.8 billion yuan, which is more than Wu's results (463.4 billion yuan), and the economic loss of this model from 2008 to 2012 is 709.6 billion yuan, which is closer to Sun's results (645.4 billion yuan).The reason for this difference is that although both models measured the impact of input changes on output after earthquake, Wu's ARIO model (2012) and Sun's Harrod-Domar model (2011) predicted the output based on government public expenditure, while this paper's model measured the output based on the total investment in society which could be reflected by night lights. Generally, the indirect economic losses assessment results in this paper are close to most of the evaluation results in other papers (Figure 13).

References

Kenneth P., John P. R.: Sichuan earthquake of 2008 [EB/OL],available at: https://www.britannica.com/event/Sichuan-earthquake-of-2008,last access:10 September 2018.

Gu L.S.: Ten Years of Wenchuan Earthquake: China's "Reconstruction Better" Program [N].South review weekly,2018-05-13(A13).

Yang S.J., The Basic Completion of Reconstruction Tasks [EB/OL],available at: http://www.gov.cn/jrzg/2010-10/25/content_1729845.htm, last access:10 September 2018

[Figure]

**Fig. 1.**

[Figure]

**Fig. 2.**

---

## Referee Comment (RC3) · Anonymous Referee #2 · 9 Oct 2018

I agree with most of the comments raised by the other referee. More specifically, I agree that you need to add to your introduction additional information about the earthquake and why that specific earthquake is of particular interest.

In addition, the use of night-time light as a proxy of economic activity should be better motivated, also linking on the existing literature. For example, see Chen X., Nordhaus W. D. (2011) Using luminosity data as a proxy for economic Statistics, Proceedings of the National Academy of Sciences, 108(21), pp. 8589–8594.

Finally, your analysis does not account for the likely increase in economic activity in areas close to the event area as a consequence of the earthquake. It could be interesting to discuss this issue.

---

## Author Comment (AC2) · 21 Oct 2018

Thanks for your contribution to this paper. I have read your comments very seriously and responded as follows:

Reply to the Comment 1: I have added the important impact of this earthquake and why was it worth studying, in expert's reply. These will be reflected in the next revision.

Reply to the Comment 2: According to your suggestion, we will add some research, focusing on the relationship between night-time light and economic activity, in the method

part. As follows:

The seismic direct economic loss refers to the damage of existing production materials and environment by earthquake, which mainly reflects the impact of earthquake disasters on economic stock. However, the seismic indirect economic loss is a systematic manifestation of losses in the chain of economic activities, which focuses on the far-reaching impact of disasters on economic flows. Studies show that night-time light can be used as a proxy of economic activity. Chen et al.(2011) had proven nighttime luminosity could be used to improve estimates of output at the regional level, Bruederle et al.(2018) conclude that nighttime lights are a good proxy for human development at the local level, Ma et al.(2014) had proved that nightlight data could be indicative of demographic and socioeconomic dynamics in China's cities. Therefore, this paper holds that the changes in night-time light after the earthquake can reflect changes in the regional economic system. The technical route is shown in Figure 3:

Reply to the Comment 3: According to your suggestion, we have added experiment module of the relationship between night-time light and economic activity. There are some interesting results will be updated to the discussion part. As follows:

The path of economic change in the disaster area after the earthquake gradually spread from the northeast to the southeast (Figure 15). The economy level for 5 years after the earthquake in disaster area (2008-2012) were compared to the level of 2007, we found that in the first year (Fig. 15-a), the economic decline is mainly concentrated in the disaster area, and the city's economic decline is most significant. In the second year (Figure 15-b), the economic slowdown area began to spread to some major cities in the southeast of the disaster area. It was not until the third year (Fig. 15-c) that the disaster area began to recover, and the first one was nat the area where the economy was reduced seriously after the disaster, but some cities located near these areas in the northeast. The force gradually spread from north to south during 2010-2012 (Fig. 15-c, Fig. 15-d, Fig. 15-e). Eventually, some cities were developed farther outside the disaster area (Figure 15-f). This explains the path of the Chinese government's

aid funds from the north to the southwest and its radiation effects on the surrounding areas.

The areas where the economy first turned around were along the road or around the city in the disaster area (Figure 16). Previous analysis have proved that the economic recovery in disaster area began around 2010. Therefore, the process of economic recovery can be studied by comparing the economic level of the disaster area from 2009 to 2012 with the level of 2008. In 2009 (Figure 16-a), excepted few areas along several roads close to the disaster area, there were few economic growth in the areas. However, in 2010 (Fig. 16-b), the economy where along most roads showed significant growth, and the surrounding areas of cities, which were seriously affected by earthquake, showed an increasing trend. By 2011 (Fig. 16-c), the trend of economic growth in the disaster areas has spread to neighboring cities and provinces, and the urban economy has also begun to recover from the periphery to center. As of 2012(Fig. 16-d), the economy of disaster area showed an interesting development pattern. On the one hand, there were few recovery in the urban economy with severe recession; on the other, a series of significant growth were shown in the surrounding areas. This may help us to distinguish the industrial layout of the disaster-stricken areas after the disaster. The economically active areas after earthquake are mainly the secondary industries such as construction and manufacturing, while those areas with slow recovery are mainly the tertiary industry such as service and entertainment industry.

References:

Chen X. ,Nordhaus W.D. :Using luminosity data as a proxy for economic statistics. Proceedings of the National Academy of Sciences, 108(21),8589–8594,2011.

Bruederle A, Hodler R. :Nighttime lights as a proxy for human development at the local level. Public Library of Science (one),13(9),e0202231,2018.

Ma T.,Zhou C.H.,Pei T., Haynie S., Fan J.F.: Responses of Suomi-NPP VIIRS-derived nighttime lights to socioeconomic activity in China's cities, Remote Sensing

Letters,5(2),165-174,2014.

[Figure]

**Figure 15:The economic change path in disaster area**
(a.The path of 2008; b.The path of 2009; c.The path of 2010;
d.The path of 2011; e.The path of 2012)

**Fig. 1.**

[Figure]

**Figure 16: Economic recovery process in the disaster area**
(a.Economic recovery area in 2009; b,Economic recovery area in 2010;
c.Economic recovery area in 2011; d.Economic recovery area in 2012)

**Fig. 2.**

---

## Author Response (AR1)

**1、Reply to the comment 1:**

**1.1 Comments from Referees**

Citation from page 3 paragraph 5: "Although it has been widely proven that there is a close relationship between night-time light and economic activity in the disaster area, due to the joint effect of economic suppression, which was caused by the earthquake damage, and economic promotion, which was caused by the reconstruction of the disaster area, the quantitative relationship between night-time light and economic statistics in the post-earthquake years is abnormal" Why is the relationship abnormal? Please explain better your point to why it is abnormal.

**1.2 Author's response**

The relationship between night-time light and economic statistics was given in Figure 5. We have a brief discussion of this issue on the page 6 paragraph 1:" Second, the largest value of Chengdu was removed from the sample, and then, the scatter plots of the two sets of data samples, which are pre-earthquake data and post-earthquake data, were compared (Figure 5). It is obtained that the correlation between the two parameters is strong in pre-earthquake data and poor in post-earthquake data." In the next revision, we will further supplement the description of the issue based on your comments.

**1.3 Author's changes in manuscript.**

Remove the following from the introduction part:

*due to the joint effect of economic suppression, which was caused by the earthquake damage, and economic promotion, which was caused by the reconstruction of the disaster area, the quantitative relationship between night-time light and economic statistics in the post-earthquake years is abnormal.*

The description of this anomaly was moved from the introduction to discussion part (line 27 of page 7):

*The quantitative relationship between night-time light and economic statistics in the post-earthquake years is abnormal. It's the result of the joint action of economic suppression, which caused by the earthquake damage, and economic promotion, which caused by the reconstruction of the disaster area.*

**2、Reply to the comment 2:**

**2.1 Comments from Referees**

Comment 2 Citation from page 3 paragraph 10: "It is very important for recovery and reconstruction to understand the indirect economic loss assessment and recovery assessment of the Wenchuan 8.0 Ms earthquake, which is a significant earthquake in recent years." Why is this specific earthquake important to investigate? The reader needs more information about the earthquake. The authors need to put this specific earthquake in a better context.

**2.2 Author's response**

The 2008 Wenchuan earthquake(Mw8.0) is the biggest earthquake event in China since 1970s. This

earthquake struck Sichuan Province, China on May 12, 2008. It killed nearly 70,000 people, 18,000 people were missing, and more than 370,000 people were injured. Millions of people were made homeless by the quake, the cost of which was estimated at $86 billion (Kenneth et al.,2013). In the three years after the disaster, Chinese government spent 865.8 billion yuan to complete 29,692 aid projects, which has brought Chinese Power to the attention of the world(Gu, 2018).

**2.3 Author's changes in manuscript.**

The contents of the response have been updated to introduction part (line 1-5 of page 2):

**3、Reply to the comment 3:**

**3.1 Comments from Referees**

Comment 3 This comment is rather general and it refers to the modelling (at page 4, 3. Method 3.2 Economic recovery evaluation model and 3.3 Economic expansion evaluation model) used to assess the economic recovery. Does the models take both increasing light intensity at one certain place and secondly, the increased spread of light i.e. new light sources at places where there were no light prior to the earthquake? Please clarify this in simple terms for the reader.

**3.2 Author's response**

In order to ensure the fairness of disaster relief assistance, government should avoid the problem that the benefits of developed regions will cover up the economic difficulties of backward regions by transfer payment system. This paper defines the light recovery and lighting expansion in the disaster area as two different processes. The 3.2 Economic recovery evaluation model only take increasing light intensity of the disaster area with light before the earthquake (Eq.6 & Eq.7). And the 3.3 Economic expansion evaluation model only take increasing light intensity of the disaster area without light before the earthquake (Eq.8)

**3.3 Author's changes in manuscript.**

The method part (line 7-20 of page 4) has been updated based on the reply content. The updated contents are as follows:

*The seismic direct economic loss refers to the damage of existing production materials and environment by earthquake, which mainly reflects the impact of earthquake disasters on economic stock. However, the seismic indirect economic loss is a systematic manifestation of losses in the chain of economic activities, which focuses on the far-reaching impact of disasters on economic flows. On the one hand, there are so many studies show that night-time light can be used as a proxy of economic activity. Chen et al.(2011) had proven nighttime luminosity could be used to improve estimates of output at the regional level, Bruederle et al.(2018) conclude that nighttime lights are a good proxy for human development at the local level, Ma et al.(2014) had proved that nightlight data could be indicative of demographic and socioeconomic dynamics in China's cities. Therefore, this paper holds that the changes in night-time light after the earthquake can reflect changes in the regional economic system. On the other hand, in order to ensure the fairness of disaster relief assistance, government should avoid the problem that the benefits of developed regions will cover up the economic difficulties of backward regions by transfer payment system. This paper defines the*

*light recovery and lighting expansion in the disaster area as two different processes. The Economic recovery evaluation model only take increasing light intensity of the disaster area with light before the earthquake. And the Economic expansion evaluation model only take increasing light intensity of the disaster area without light before the earthquake. The technical flowchart is shown in Figure 3:*

**4、Reply to the comment 4 :**

**4.1 Comments from Referees**

Comment 4 4.2 Economic recovery progress in Sichuan Province. Please start the paragraph with for example: Our results indicate that it has taken more than 5 years…Now the paragraph can be interpreted as a common observation. Make sure that this type of statement are done in line with your empirical results.

**4.2 Author's response**

Our results indicate that the economic recovery in Sichuan Province took more than 5 years, which is far more than the time spent on reconstruction (Yang,2012).

**4.3 Author's changes in manuscript.**

The contents of the response have been updated to the first sentence of *Economic recovery progress in Sichuan Province* part (line17 of page 6):

**5、Reply to the comment 5 :**

**5.1 Comments from Referees**

Comment 5 5 Discussion Citation from Page 7 Paragraph 5-10. "When considering the short-term impact of the stoppage of production after the earthquake, the indirect economic loss in 2008 evaluated by Lu (2008) was 168 billion yuan, which is 176% of the results found in this paper. According to the changes of economic growth rate after the earthquake, Sun (2011) predicted that the indirect loss after the earthquake in 2008 was 100.8 billion yuan, which is 105% of the results found in this paper." Please clarify this statement. What is the 176% resp. 105% telling us? Are your model overestimating the recovery? This is unclear in the present format.

**5.2 Author's response**

Compared with statistical data, night-time light can not only reflect the spatial difference of economic expansion and recovery but also guarantee the accuracy of assessment. In terms of short-term economic loss, the economic loss of this model in 2008 is 95.7 billion yuan, which is closer to Sun's results (100.8 billion yuan), but lower than Lu's results (168 billion yuan). The reason for this difference is that although these models taken into account the capacity reduction caused by earthquake, Lu's model (2008) assumed that the affected department was completely shut down during a period of time, while this paper's model and Sun's model (2011) assumed that the production capacity was gradually restored. In terms of mid-long term economic loss, the economic loss of this model from 2008 to 2011 is 596.8 billion yuan, which is more than Wu's results (463.4 billion yuan), and the economic loss of this model from 2008 to 2012 is 709.6 billion yuan, which

is closer to Sun's results (645.4 billion yuan).The reason for this difference is that although both models measured the impact of input changes on output after earthquake, Wu's ARIO model (2012) and Sun's Harrod-Domar model (2011) predicted the output based on government public expenditure, while this paper's model measured the output based on the total investment in society which could be reflected by night lights. Generally, the indirect economic losses assessment results in this paper are close to most of the evaluation results in other papers (Figure 13).

**5.3 Author's changes in manuscript.**

The contents of the response have been updated to the discussion part (line 14-26 of page 7):

**6、Reply to the comment 6 :**

**6.1 Comments from Referees**

In addition, the use of night-time light as a proxy of economic activity should be better motivated, also linking on the existing literature. For example, see Chen X., Nordhaus W. D. (2011) Using luminosity data as a proxy for economic Statistics, Proceedings of the National Academy of Sciences, 108(21), pp. 8589–8594

**6.2 Author's response**

According to your suggestion, we will add some research, focusing on the relationship between night-time light and economic activity, in the method part. As follows:

*The seismic direct economic loss refers to the damage of existing production materials and environment by earthquake, which mainly reflects the impact of earthquake disasters on economic stock. However, the seismic indirect economic loss is a systematic manifestation of losses in the chain of economic activities, which focuses on the far-reaching impact of disasters on economic flows. Studies show that night-time light can be used as a proxy of economic activity. Chen et al.(2011) had proven nighttime luminosity could be used to improve estimates of output at the regional level, Bruederle et al.(2018) conclude that nighttime lights are a good proxy for human development at the local level, Ma et al.(2014) had proved that nightlight data could be indicative of demographic and socioeconomic dynamics in China's cities. Therefore, this paper holds that the changes in night-time light after the earthquake can reflect changes in the regional economic system. The technical route is shown in Figure 3:*

**6.3 Author's changes in manuscript.**

This change is same as *Reply to the comment 3.*

**7、Reply to the comment 7 :**

**7.1 Comments from Referees**

Finally, your analysis does not account for the likely increase in economic activity in areas close to the event area as a consequence of the earthquake. It could be interesting to discuss this issue.

**7.2 Author's response**

According to your suggestion, we have added experiment module of the relationship between night-time light and economic activity. There are some interesting results will be updated to the discussion

part. As follows:

*The path of economic change in the disaster area after the earthquake gradually spread from the northeast to the southeast (Figure 15). The economy level for 5 years after the earthquake in disaster area (2008-2012) were compared to the level of 2007, we found that in the first year (Fig. 15-a), the economic decline is mainly concentrated in the disaster area, and the city's economic decline is most significant. In the second year (Figure 15-b), the economic slowdown area began to spread to some major cities in the southeast of the disaster area. It was not until the third year (Fig. 15-c) that the disaster area began to recover, and the first one was not the area where the economy was reduced seriously after the disaster, but some cities located near these areas in the northeast. The force gradually spread from north to south during 2010-2012 (Fig. 15-c, Fig. 15-d, Fig. 15-e). Eventually, some cities were developed farther outside the disaster area (Figure 15-f). This explains the path of the Chinese government's aid funds from the north to the southwest and its radiation effects on the surrounding areas.*

*The areas where the economy first turned around were along the road or around the city in the disaster area (Figure 16). Previous analysis have proved that the economic recovery in disaster area began around 2010. Therefore, the process of economic recovery can be studied by comparing the economic level of the disaster area from 2009 to 2012 with the level of 2008. In 2009 (Figure 16-a), excepted few areas along several roads close to the disaster area, there were few economic growth in the areas. However, in 2010 (Fig. 16-b), the economy where along most roads showed significant growth, and the surrounding areas of cities, which were seriously affected by earthquake, showed an increasing trend. By 2011 (Fig. 16-c), the trend of economic growth in the disaster areas has spread to neighboring cities and provinces, and the urban economy has also begun to recover from the periphery to center. As of 2012(Fig. 16-d), the economy of disaster area showed an interesting development pattern. On the one hand, there were few recovery in the urban economy with severe recession; on the other, a series of significant growth were shown in the surrounding areas. This may help us to distinguish the industrial layout of the disaster-stricken areas after the disaster. The economically active areas after earthquake are mainly the secondary industries such as construction and manufacturing, while those areas with slow recovery are mainly the tertiary industry such as service and entertainment industry.*

**7.3 Author's changes in manuscript.**

The contents of the response have been updated to the discussion part (line 13-33 of page8):

**8、Other changes in manuscript:**

**8.1 The change in conclusion part**

Updated the conclusion based on the new findings of the discussion part (line 2-19 of page9):

[revised manuscript text omitted]

批注 [王建飞2]: Reply to the comment 1：
The description of this anomaly was moved from the introduction to discussion part

[revised manuscript text omitted]

批注 [王建飞5]: Reply to the comment 5：

批注 [王建飞6]: Reply to the comment 1：
The description of this anomaly was moved from the introduction to here

[revised manuscript text omitted]

批注 [王建飞10]: Reply to the comment 7：

---

## Author Response (AR2)

**Dear editorial:**

**First, according to your comments, i have finished the revisions point-by-point. Second, we also revised the figures according to the description of the paper. Third, according to your suggestion, this paper has been edited by an English-speaking person. The marked-up manuscript, which highlights the changes, is as follows:**

[revised manuscript text omitted]

**Figure 2 Normalized total lightness of uncorrected and corrected data.**

[Figure]

**Figure 3: Technical flowchart.**

[Figure]

**Figure 4: Correlation coefficient between GDP and DN-related parameters.**

[Figure]

**Figure 5:** DN-GDP scatter plots of pre- and post-earthquakes in Sichuan (Maxima have been excluded).

[Figure]

$$y = 4E\text{-}08x^2 + 0.0042x + 96.285$$
$$R^2 = 0.9293$$

5          **Figure 6:** Total DN and GDP Regression function of Pre-earthquake in Sichuan .

[Figure]

**Figure 7: The time of light recovery**

[Figure]

**Figure 8: The poportion of light recovery area.**

[Figure]

[Figure]

**Figure 9:** The ratio of the post-earthquake DN to pre-earthquake in Sichuan (DN_Ratio)

[Figure]

**Figure 10: Distribution of new lighting areas after earthquake**

[Figure]

**Figure 11: The**  cumulative DN of expansion area

[Figure]

[Figure]

**Figure 12: The** average DN in expansion area

[Figure]

**Figure 13: Comparison of indirect loss with other research**

[Figure]

[Figure]

**Figure 14: The economic change path in disaster area ; (a) The path of 2008; (b) The path of 2009; (c) The path of 2010; (d) The path of 2011; (e) The path of 2012;**

[Figure]

[Figure]

**Figure 15: Economic recovery process in the disaster area ; (a) Economic recovery area in 2009; (b) Economic recovery area in 2010; (c) Economic recovery area in 2011; (d) Economic recovery area in 2012;**